# Base editing in bovine embryos reveals a species-specific role of SOX2 in regulation of pluripotency

**Lei Luo**[1⦿], **Yan Shi**[1⦿], **Huanan Wang**[1,2⦿], **Zizengchen Wang**[1,2], **Yanna Dang**[1], **Shuang Li**[1], **Shaohua Wang**[1], **Kun Zhang**[1]*

**1** Laboratory of Mammalian Molecular Embryology, Department of Animal Science and Technology, College of Animal Sciences, Zhejiang University, Hangzhou, China, **2** Department of Veterinary Science, College of Animal Sciences, Zhejiang University, Hangzhou, China

⦿ These authors contributed equally to this work.
* kzhang@zju.edu.cn

**Data Availability Statement:** All RNA-seq files are available from the Gene Expression Omnibus database (accession number GSE189318).

## Abstract

The emergence of the first three lineages during development is orchestrated by a network of transcription factors, which are best characterized in mice. However, the role and regulation of these factors are not completely conserved in other mammals, including human and cattle. Here, we establish a gene inactivation system with a robust efficiency by introducing premature codon with cytosine base editors in bovine early embryos. By using this approach, we have determined the functional consequences of three critical lineage-specific genes (*SOX2*, *OCT4* and *CDX2*) in bovine embryos. In particular, *SOX2* knockout results in a failure of the establishment of pluripotency in blastocysts. Indeed, OCT4 level is significantly reduced and NANOG barely detectable. Furthermore, the formation of primitive endoderm is compromised with few SOX17 positive cells. RNA-seq analysis of single blastocysts (day 7.5) reveals dysregulation of 2074 genes, among which 90% are up-regulated in *SOX2*-null blastocysts. Intriguingly, more than a dozen lineage-specific genes, including *OCT4* and *NANOG*, are down-regulated. Moreover, SOX2 level is sustained in the trophectoderm in absence of CDX2. However, *OCT4* knockout does not affect the expression of SOX2. Overall, we propose that SOX2 is indispensable for OCT4 and NANOG expression and CDX2 represses the expression of SOX2 in the trophectoderm in cattle, which are all in sharp contrast with results in mice.

## Author summary

The first and second cell fate decisions of a new life are important for subsequent embryonic and placental development. These events are finely controlled by a network of transcriptional factors, which are extensively characterized in mice. Species-specific roles of these proteins are emerging in mammals. Here, we develop a gene loss-of-function system by using cytosine base editors in bovine early embryos. We find that expression pattern, functional roles, and regulation of SOX2 are all different between mouse and bovine

**Funding:** KZ is supported by the National Natural Science Foundation of China (No. 31872348, No. 32072731 and No. 32161143032)(https://www.nsfc.gov.cn/). HW is funded by the National Natural Science Foundation of China (No. 32072939) (https://www.nsfc.gov.cn/). LL and SW are both supported by the National Natural Science Foundation of China (No.31941007)(https://www.nsfc.gov.cn/). KZ is also funded by Zhejiang Provincial Natural Science Foundation (LZ21C170001) (http://zjnsf.kjt.zj.gov.cn/portal/index.html). HW is also supported by Zhejiang Provincial Natural Science Foundation (No. LY19C180002)(http://zjnsf.kjt.zj.gov.cn/portal/index.html). LL is funded by China Postdoctoral Science Foundation (No. 2020M671742)(http://jj.chinapostdoctor.org.cn/website/index.html). The funders had no role in study design, data collection and analysis, decision to publish, or preparation of the manuscript.

**Competing interests:** The authors have declared that no competing interests exist.

embryos. Remarkably, SOX2 is extremely important for OCT4 and NANOG, two well-established pluripotency factors. Furthermore, CDX2 is required to repress SOX2 in the trophectoderm. Given similar expression pattern of SOX2 between human and bovine blastocysts, bovine embryo represents a putative model to investigate human pluripotency regulation in vivo.

## Introduction

Mammalian preimplantation development is characterized by the first sequential cell fate decisions occurring in close temporal relationship to each other. The first cell fate decision gives rise to the inner cell mass (ICM) and the trophectoderm (TE) and the ICM subsequently generates the primitive endoderm (PE) and the epiblast (EPI) during the second cell fate decision. TE and PE will develop into placenta and extra-embryonic cells, respectively, whereas the pluripotent EPI will contribute to the embryo proper [1,2]. The mechanisms that regulate these events have been mostly obtained from mouse model. Recent gene-expression and functional analyses suggest that these mechanisms in the mouse may differ in other mammals, including human and cattle [3–7]. Investigation of these mechanisms is important for assisted reproductive technology, regenerative medicine as well as understanding early embryonic mortality in humans and agricultural animals.

The establishment and maintenance of pluripotency are regulated by a variety of transcription factors, including core pluripotency factors, OCT4, SOX2 and NANOG [1,2]. The functional significance and relationship of these factors have been relatively well-characterized in mouse embryos. Interestingly, unlike the universal expression pattern of OCT4 at morula stage, SOX2 is specifically restricted into the inside cells of morula that become the ICM and is considered the earliest pluripotency marker in mice [3]. The HIPPO pathway plays a critical role in the temporal and spatial expression of SOX2 in mouse preimplantation embryos [3,8]. However, SOX2 is not required for the first cell fate decision although *SOX2*-null mouse embryos die soon after implantation and exhibit abnormal ICM [9]. Meanwhile, SOX2 is dispensable for the initial expression of OCT4 and NANOG in mouse blastocysts [3]. In contrast with the expression pattern in mouse preimplantation embryos, SOX2 is not restricted until the expanded blastocyst stage in cattle and humans [10], suggesting a differential regulation of pluripotency in these species.

Base editors are derived from CRISPR/Cas9 genome-editing system and used for precise base editing without DNA double-strand breaks and homology-directed repair [11]. Adenine base editors (ABEs) are used to convert A:T base pairs to a G:C base pairs [12]. Cytosine base editors are used to convert C:G to T:A [13]. In addition, a specific function of cytosine base editors is to install a premature stop codon by converting the four codons CAA, CAG, CGA and TGG into stop codons TAA, TAG or TGA [14,15]. To date, base editing has been successfully implemented in the embryos of mice [16,17], rats [18], pigs [19], rabbits [20], cynomolgus monkeys [21], and humans [22–24] but has yet been determined in cattle.

In the present study, we successfully develop a highly-efficient base editing system in bovine embryos, representing a powerful tool to interrogate gene functions. We then address the role of SOX2 in bovine early embryonic development. *SOX2*-null embryos can develop to blastocyst stage, but the ICM is abnormal. *SOX2* knockout (KO) results in a significant reduction in OCT4 and NANOG expression as well as dysregulated expression of over 2000 genes in bovine blastocysts. Impressively, CDX2 inhibits SOX2 expression in the TE. In summary, SOX2 is

**Table 1. Results of multiple genes' editing in bovine embryos by BE3.**

| No. of embryos sequenced | No. of unedited (%) | No. of single gene-edited (%) | | | No. of two genes-edited (%) | | | No. of three genes-edited (%) |
|---|---|---|---|---|---|---|---|---|
| | | *SMAD4* | *TEAD4* | *CDX2* | *SMAD4* and *TEAD4* | *TEAD4* and *CDX2* | *SMAD4* and *CDX2* | *SMAD4, TEAD4* and *CDX2* |
| 31 | 11 (33.5) | 2 (6.5) | 1 (3.2) | 1 (3.2) | 0 (0.0) | 1 (3.2) | 7 (22.6) | 8 (25.8) |

important for OCT4 and NANOG expression and CDX2 is necessary for ensuring the restricted expression of SOX2 in the ICM of bovine blastocysts.

# Results

## Base editors enable efficient genome editing in bovine embryos

We first sought to establish base editing system using cytosine base editor 3 (BE3) and adenine base editor 7.10 (ABE7.10). Base editor mRNA and sgRNAs targeting *SMAD4* were co-injected into bovine zygotes. Morula cultured in vitro for 6 days (D6) were collected and genotypes identified by both Sanger sequencing and targeted deep sequencing (S1A and S1B Fig). As for BE3, the desired mutations of $C_6$ and $C_7$ to T were found in all embryos examined (S1C Fig). The average efficiency of $C_6$ and $C_7$ being edited as T was 86.3% and 85.4%, respectively, compared with 5.0% in wildtype (WT) embryos (S1D Fig). Regarding ABE7.10, the target mutation of $A_5$ to G was also found in all embryos with an average editing efficiency of 79.4% versus 4.5% in WT embryos (S1E and S1F Fig). In addition, we investigated the off-target effect by targeted next-generation sequencing and found no obvious distal off-target edits at the six predicted off-target sites (S2A and S2B Fig).

To evaluate the ability of base editors to edit multiple genes in bovine embryos, we simultaneously injected sgRNAs targeting three genes, *SMAD4, TEAD4*, and *CDX2*, with base editor mRNA. Results show that injection of the base editing components did not affect the embryonic development to D6 morula(S3A and S3B Fig). Using BE3, results indicate successful editing of three, two and single target genes in 25.8%, 25.8% and 12.9% embryos, respectively (S3C Fig and Table 1). For ABE7.10, results indicate successful editing of three, two and single target genes in 23.3%, 56.7% and 20.0% embryos, respectively (S3D Fig and Table 2). Moreover, the percentages of editing for *SMAD4, TEAD4* and *CDX2* by BE3 is 54.8%, 32.3% and 54.8%, respectively and those by ABE7.10 are 76.7%, 33.3% and 93.3%, respectively. Taken together, these data present proof-of-evidence of base editing with high efficiency in bovine embryos.

## Disrupting genes by introducing premature stop codon with cytosine base editors in bovine embryos

Cytosine base editors can introduce premature stop codons to inactivate genes by precisely converting four codons into stop codons [14,15]. We next tested the feasibility of disrupting a

**Table 2. Results of multiple genes' editing in bovine embryos by ABE7.10.**

| No. embryos sequenced | No. unedited (%) | No. of single gene-edited (%) | | | No. of two genes-edited (%) | | | No. of three genes-edited (%) |
|---|---|---|---|---|---|---|---|---|
| | | *SMAD4* | *TEAD4* | *CDX2* | *SMAD4* and *TEAD4* | *TEAD4* and *CDX2* | *SMAD4* and *CDX2* | *SMAD4, TEAD4* and *CDX2* |
| 30 | 0 (0.0) | 0 (0.0) | 1 (3.3) | 5 (16.7) | 1 (3.3) | 1 (3.3) | 15 (50.0) | 7 (23.3) |

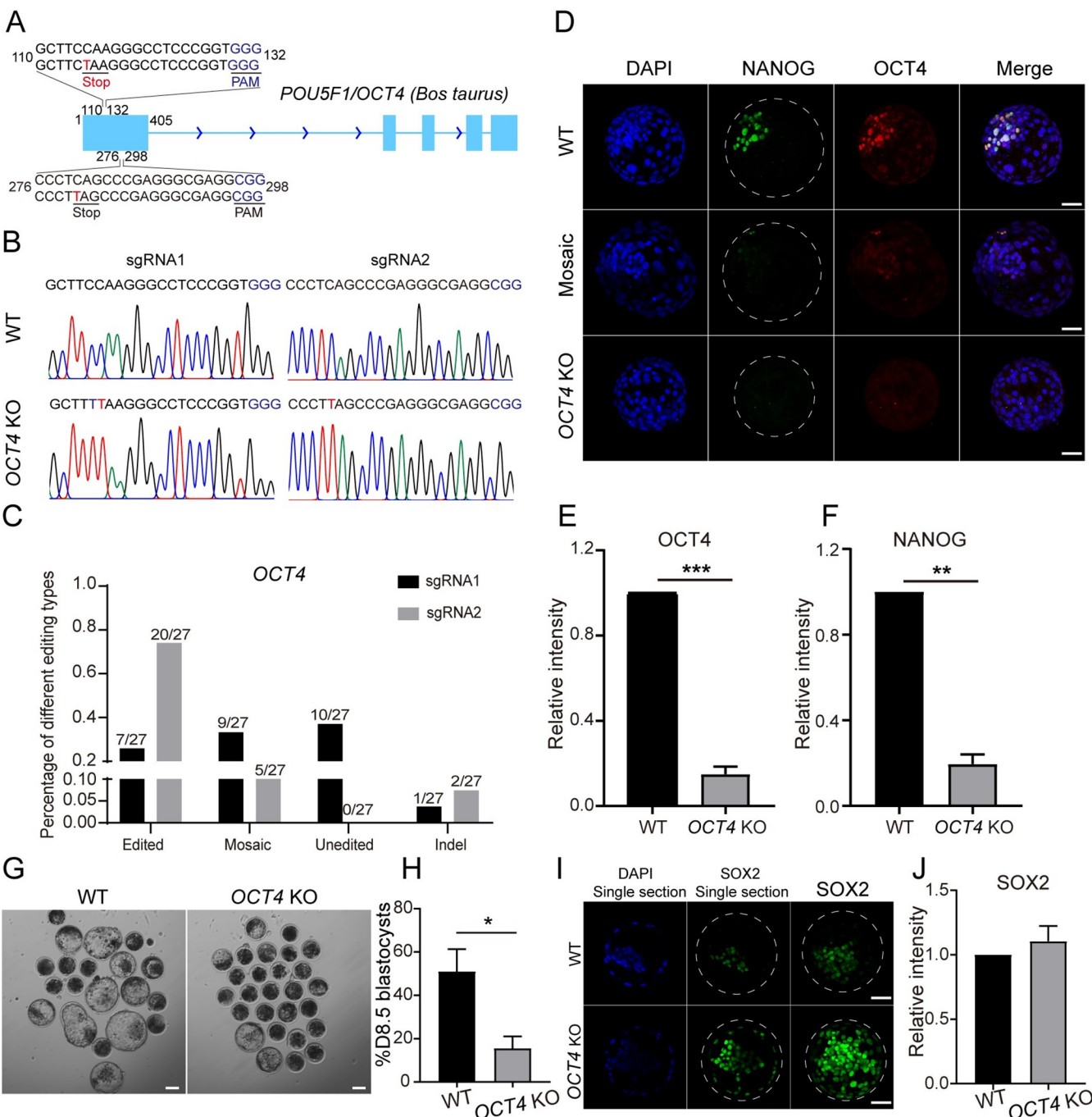

**Fig 1. Effects of *OCT4* knockout on bovine early embryonic development.** A. Two sgRNAs designed to target *OCT4*. The red letters represent potential editing sites. The blue letters represent PAM sequences. B. Representative Sanger sequencing results. The red letters represent editing sites. C. Editing types of *OCT4* sgRNA1 and sgRNA2 (27 embryos were analyzed). D-F. Immunostaining detection of OCT4 and NANOG in D8.5 WT and *OCT4* KO groups (Two replicates of 4–6 blastocysts per group). Green: NANOG; Red: OCT4. Scale bar = 50 μm. G and H: *OCT4* KO results in the decrease of blastocyst rate at D8.5 (Three replicates of 20–25 embryos per group). Scale bar = 100 μm. I and J. Immunostaining analysis of SOX2 expression and distribution in D8.5 blastocysts (Two replicates of 4–6 blastocysts per group). Green: SOX2; Red: OCT4. Scale bar = 50 μm.

gene in bovine early embryos. To maximize the editing efficiency, we designed and co-injected 2 sgRNAs targeting *OCT4* (Fig 1A). Results show that the edited efficiency of sgRNA1 is 25.92% and the one of sgRNA2 reaches 74.07% (Fig 1B and 1C). Overall, premature stop

codons were successfully introduced in 77.8% (21 out of 27) embryos. As a side-by-side experiment, immunostaining analysis confirms that OCT4 can be efficiently deleted in all blastomeres in D8.5 blastocysts (Fig 1D and 1E). Next, we tested if *OCT4*-null embryos generated here recapitulated the phenotype of *OCT4* KO embryos produced via somatic cell nuclear transfer as reported previously [6]. Similarly, NANOG is barely detectable in absence of OCT4 in D8.5 blastocysts (Fig 1D and 1F). The developmental potential to form blastocysts is greatly inhibited in *OCT4* KO groups (Fig 1G and 1H). Moreover, there is no significant difference in the signal intensity of SOX2 for SOX2 positive cells between control and *OCT4* KO groups. However, SOX2 is sustained in the TE cells, which could be attributed to the developmental delay of D8.5 *OCT4* KO blastocysts (Fig 1I and 1J). Thus, we establish a powerful and reliable system to accomplish efficient base editing in bovine embryos, which will facilitate studies of gene functions in bovine embryos.

## Expression pattern of SOX2 protein in bovine early embryos

To functionally characterize SOX2 in bovine embryos, we first determine its expression pattern in detail in bovine embryos. SOX2 was first found at the 8-cell stage and continued to be expressed thereafter (Fig 2A). In contrast to mouse embryos, SOX2 was not detected during oocyte maturation and early development to the four-cell stage (Fig 2A). It is noteworthy that SOX2 gradually accumulates in the ICM cells along with blastocyst expansion. Specifically, SOX2 was evenly distributed in both TE and ICM in D7.5 early blastocysts. Then, SOX2 was lost in subsets of TE cells in D7.5 middle blastocysts and eventually restricted into ICM in D8.5 late blastocysts (Fig 2B), which is consistent with previous research [10]. Quantitative results show SOX2 level in TE is gradually diminished relative to the one in ICM when the blastocyst is expanding (Fig 2C and 2D). Altogether, these data indicate that SOX2 displays a different expression pattern in bovine embryos, suggesting species-specific differences in the regulatory mechanism of pluripotency.

## Effects of SOX2 KO on the bovine early embryonic development

We then sought to explore the functional role of SOX2 by disrupting its expression by using BE3 (Fig 3A and 3B). Genotyping results show that sgRNA2 and 3 are more efficient than sgRNA1 in editing SOX2 (Fig 3C and 3D). Overall, premature stop codon was successfully installed at SOX2 in 87.1% (101 out of 116) bovine blastocysts when these three sgRNAs were co-injected. Immunostaining results further confirm that SOX2 signal was drastically diminished in these corresponding blastocysts and only 9 (out of 116) embryos display mosaicism (Fig 3E).

In vitro culture of embryos reveals no significant difference in the capability to become blastocysts between *SOX2* KO and WT groups (Fig 3F). Interestingly, the total cell number per blastocyst was significantly reduced at both D7.5 and D8.5 (Fig 3G and 3H). Overall, SOX2 is not required for blastocyst formation in cattle.

## *SOX2* knockout disrupts the network of pluripotent genes in bovine blastocysts

To determine the molecular consequence of *SOX2* KO, RNA-seq of single early blastocysts was performed at D7.5. The samples are morphological indifferent upon collection to avoid bias (Figs 4A, S4A and S4B). A partial cDNA library from each blastocyst was used for genotyping (Fig 4A and S4C). Principal component analysis (PCA) reveals that WT and D7.5 *SOX2* KO blastocysts formed two distinguished clusters (S4D Fig). Results also show that the transcriptome of D7.5 SOX2 KO blastocysts cluster together with D7.5 WT blastocysts rather than

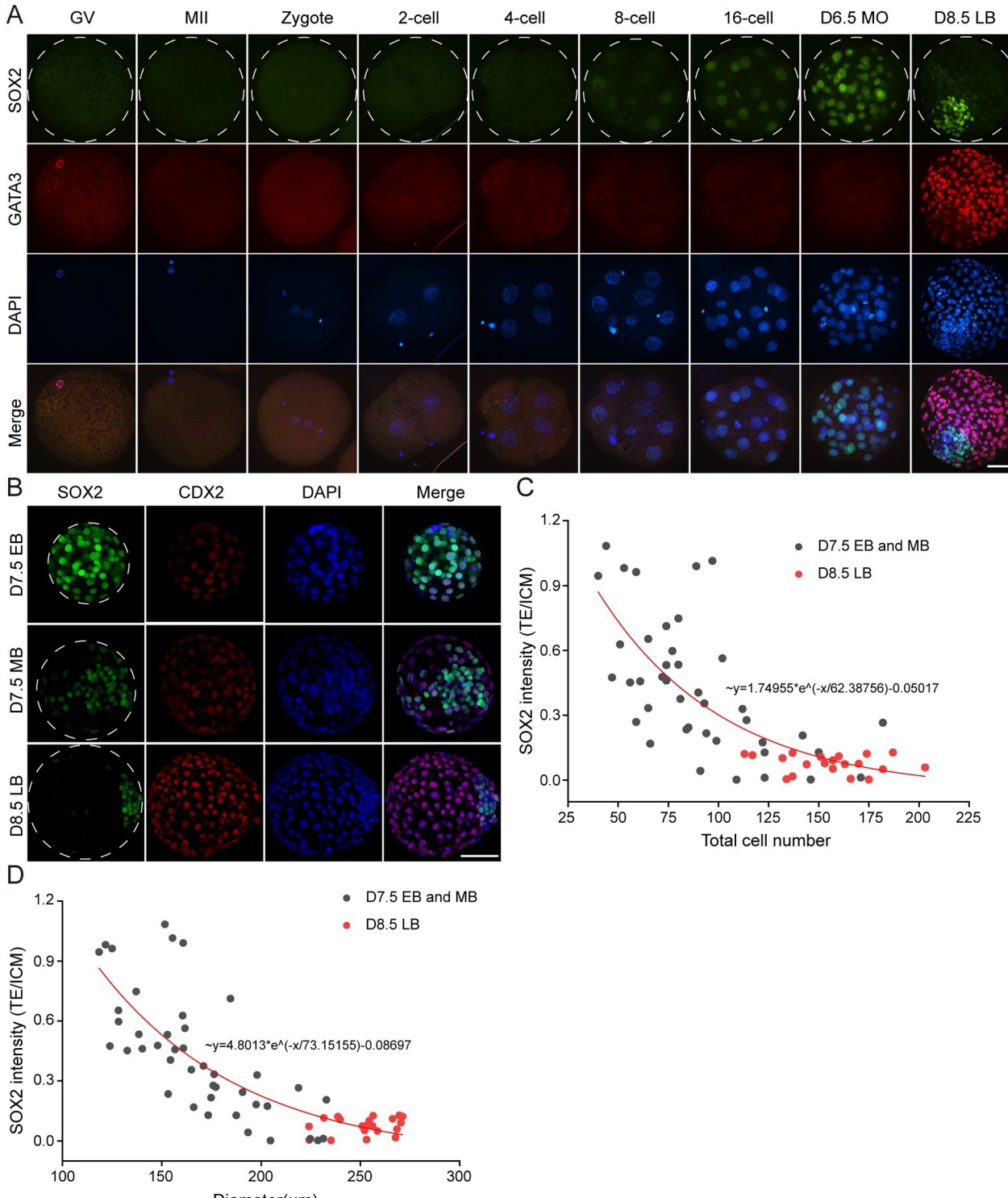

**Fig 2. Dynamic expression pattern of SOX2 during bovine early embryonic development.** A. Immunofluorescence detection of SOX2 and GATA3 during oocyte maturation and embryonic development. Green: SOX2 protein; Red: GATA3 protein; Blue: DAPI (Nuclei). The experiment was independently replicated two times with at least 10 oocytes or embryos per stage analyzed. Scale bar = 50 μm. GV: germinal vesicle, MII: metaphase II, D6.5 MO: D6.5 morula, D8.5 LB: D8.5 late blastocysts. B. Immunofluorescence analysis of SOX2 protein along with blastocyst expansion. Green: SOX2; Red: CDX2; Scale bar = 50 μm. D7.5 EB: D7.5 early blastocyst, D7.5 MB: D7.5 middle blastocyst, D8.5 LB: D8.5 late blastocyst. C and D: The correlation between SOX2 intensity (TE/ICM) and total cell number or diameter. Note: the red bullets represent D8.5 late blastocysts.

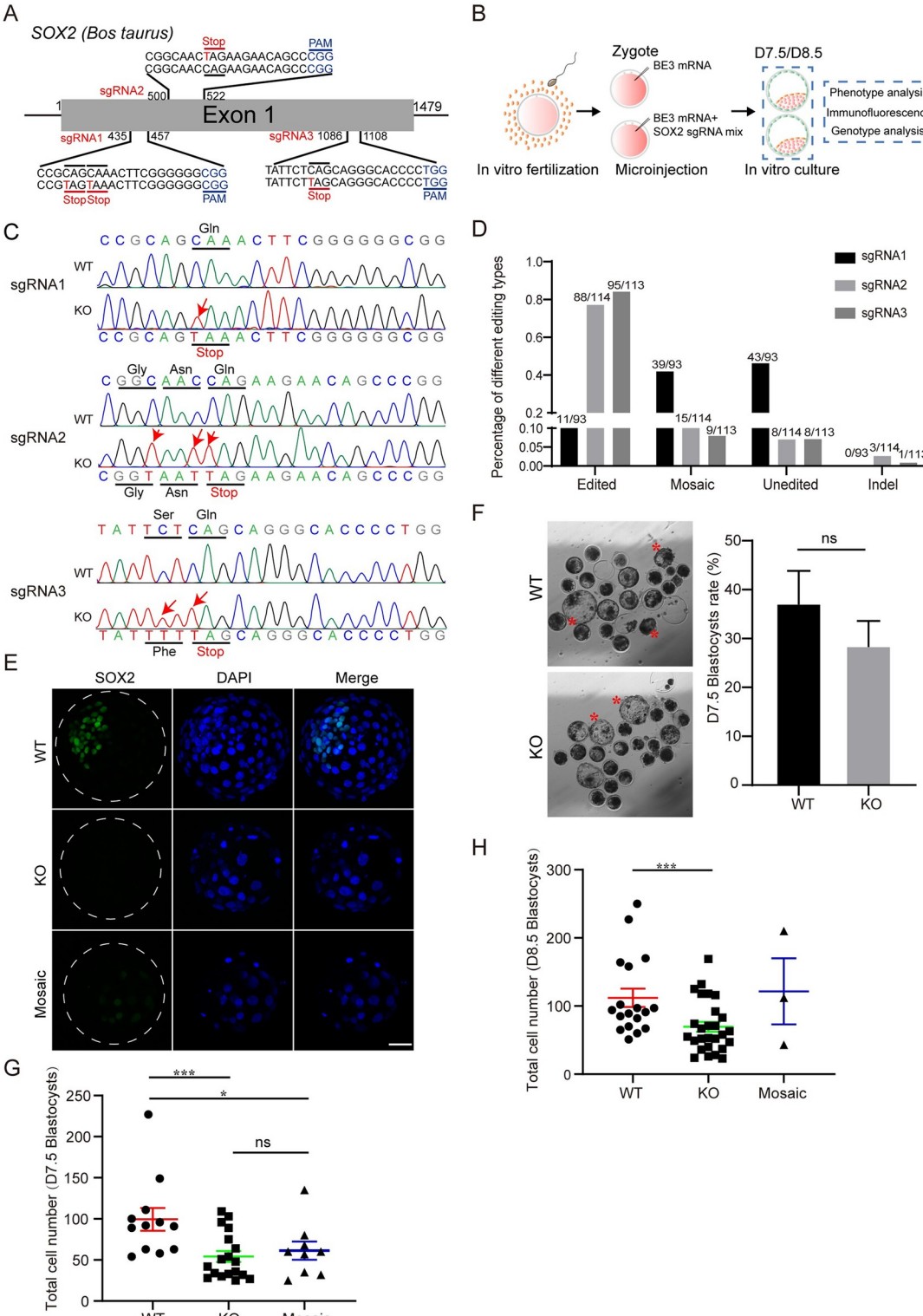

**Fig 3. Effects of *SOX2* knockout on bovine early embryonic development.** A. sgRNAs used to target *SOX2*. Red lines represent the position of introduced premature stop codon. B. Experimental design to explore the effects of *SOX2* KO on bovine early embryonic development. C. Representative genotyping results for three distinct sgRNAs. WT: wild-type; KO: putative *SOX2* knockout embryos. Red arrows denote successful C: T conversion. D. Statistical analysis of editing efficiency for each sgRNA. The target sequence of sgRNA1, sgRNA2 and sgRNA3 were analyzed in 93, 113 and 114 embryos, respectively. E.

Immunostaining validation for *SOX2* KO at D7.5 blastocyst (BL) stages (Three replicates of 3–5 embryos were analyzed per group). Scale bar = 50 μm. F. Blastocyst formation rate of bovine embryos after *SOX2* KO. The rate of blastocysts at D7.5 was recorded with no significant difference found between WT and KO groups (Five independent replicates of 20–25 embryos per group). Red asterisks represent hatching blastocysts. Scale bar = 100 μm. G and H: Statistical analysis of total cell numbers at D7.5 (G) and D8.5 (H). Asterisks refer to significant differences (*:P < 0.05; **:P <0.05; ***: P<0.001).

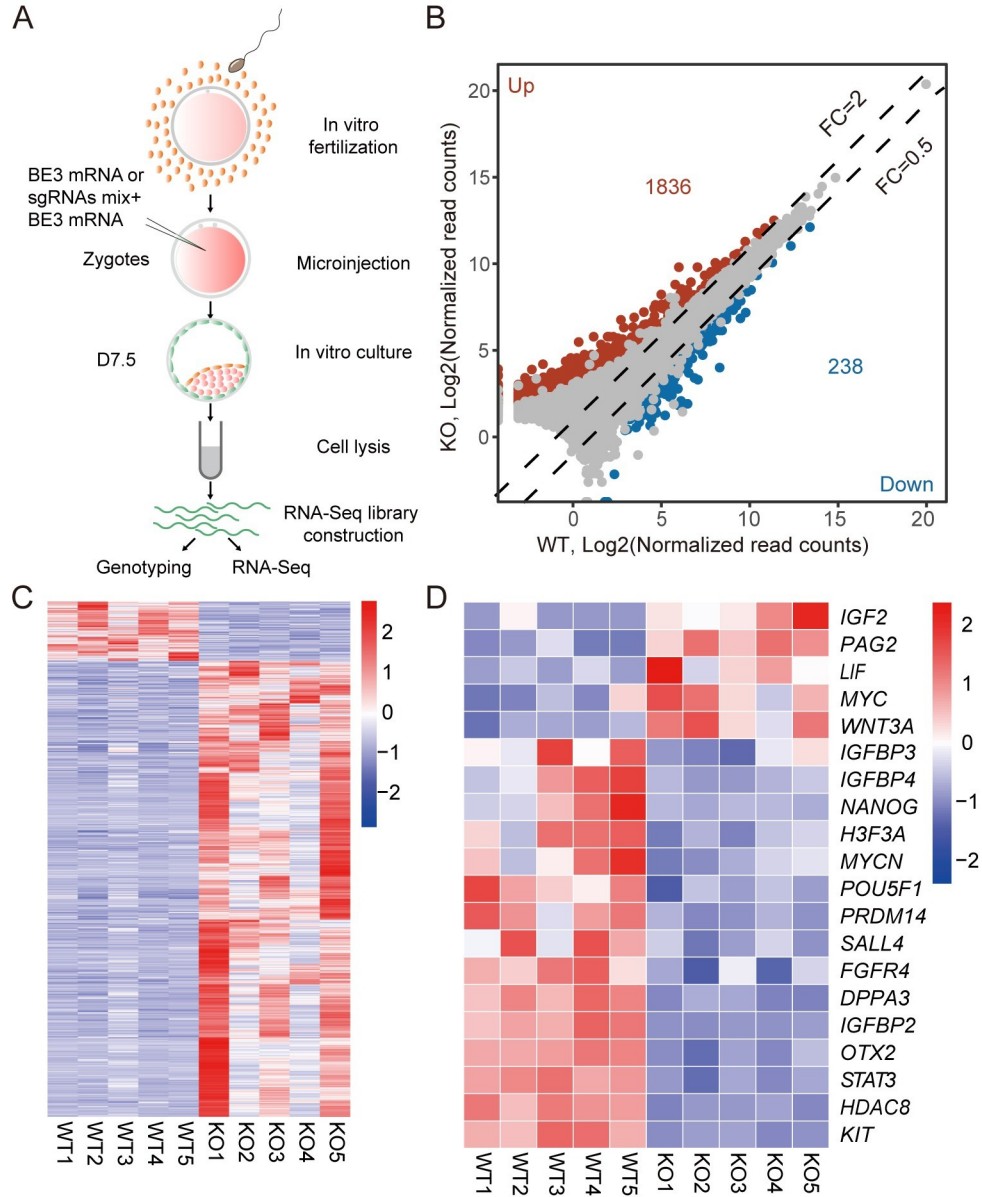

**Fig 4. *SOX2* knockout disrupts the network of pluripotency genes in bovine blastocysts.** A. Experimental scheme of single D7.5 early blastocyst RNA-seq in WT and *SOX2* KO groups. B. Volcano plot depicting differentially expressed genes, among which 1836 are upregulated and 238 are downregulated (Fold Change> = 2 or < = 0.5; Padj< = 0.05). C. Heat map showing all differentially expressed genes between WT and *SOX2* KO groups. D. Heat map showing differential expression of genes involved in the regulation of pluripotency between WT and *SOX2* KO group.

D6 WT morula (GSE158679; S4E and S4F Fig). There is a total of 2074 differentially expressed genes (DEGs, Threshold: Fold changes (FC) $> = 2$ or $< = 0.5$, False Discovery Rate (FDR) adjusted P-value $< = 0.05$), among which 88.53% were remarkably upregulated in *SOX2* KO groups (Fig 4B and 4C).

Gene ontology (GO) analysis reveals that the top GO terms enriched in DEGs include membrane depolarization during an action potential, cell adhesion, integral component of plasma membrane, calcium ion binding. Interestingly, we found that a number of overrepresented genes are involved in the regulation of the pluripotency, including up-regulated *IGF2*, *PAG2*, *LIF*, *MYC*, *WNT3A*, and down-regulated *IGFBP3*, *IGFBP4*, *PRDM14*, *SALL4*, *FGFR4*, *STAT3*, *HDAC8* (Fig 4D). Surprisingly, *NANOG* and *OCT4* were both sharply downregulated in D7.5 *SOX2* KO blastocysts. In sum, these data suggest SOX2 plays a critical role in maintaining gene expression of the pluripotency network.

## SOX2 is indispensable for NANOG and OCT4 expression in the ICM of bovine blastocysts

We then hypothesized that SOX2 is required for OCT4 and NANOG expression in bovine blastocysts. As reported previously, we confirmed that OCT4 is evenly localized in the ICM and TE at D7.5 early and middle blastocyst stage but gradually restricted into the ICM at D8.5 late blastocyst stage in cattle (S5A and S5B Fig).

Results show that the TE cell number (CDX2 positive) was not obviously changed while the number of ICM cells (CDX2 negative) was decreased dramatically in D8.5 *SOX2* KO blastocysts (Fig 5A). Remarkably, the intensity of OCT4 decreased significantly in both D7.5 and D8.5 *SOX2* KO blastocysts (Fig 5B). NANOG was first detected in D6.5 morula and distributed in both TE and ICM of D7.5 blastocysts and then fast aggregated into the EPI in D8.5 late blastocysts (S5B Fig). Intriguingly, NANOG was barely seen in both D7.5 and D8.5 *SOX2* KO blastocysts (Fig 5C). To determine if SOX2 KO affect NANOG and OCT4 at earlier stage prior to blastocyst formation, we performed IF analysis on D6.5 and found OCT4 and NANOG signal intentsity were both reduced in SOX2 KO embryos (S5C and S5D Fig). These results collectively suggest that SOX2 is required for maintaining the correct expression profile of NANOG and OCT4 in bovine blastocysts.

To further determine if *SOX2* KO affects the specification of the PE, we performed immunostaining against SOX17, an established marker of the PE, and found the number of SOX17 positive cells was greatly reduced in D8.5 *SOX2* KO blastocysts (Fig 5D), suggesting a compromised formation of the PE.

## CDX2 is required for the restricted expression of SOX2 in the ICM of bovine late blastocysts

We next asked if CDX2 was involved in the gradual disappearance of SOX2 in the TE. Both genotyping and immunostaining results indicate CDX2 is completely knocked out of 82.9% of embryos (58 out of 70; S6A–S6D Fig). No difference was found in the developmental potential to arrive blastocyst stage in *CDX2* KO groups (S6E Fig). Immunostaining analysis reveals that SOX2 signal is sustained in the TE of D8.5 *CDX2* KO blastocysts (Fig 6A and 6B). To further test the specificity of the role of CDX2 in repressing SOX2 expression during bovine embryonic development, we microinjected base editing components into one blastomere at 2-cell stage (Fig 6C). Immunostaining and confocal microscopy analysis indicates that SOX2 signal of CDX2 negative cells is obviously brighter than those of CDX2 positive cells in the TE of D8.5 mosaic blastocysts (Fig 6D and 6E), further consolidating the conclusion that CDX2 is required to diminish SOX2 in the TE of bovine blastocysts.

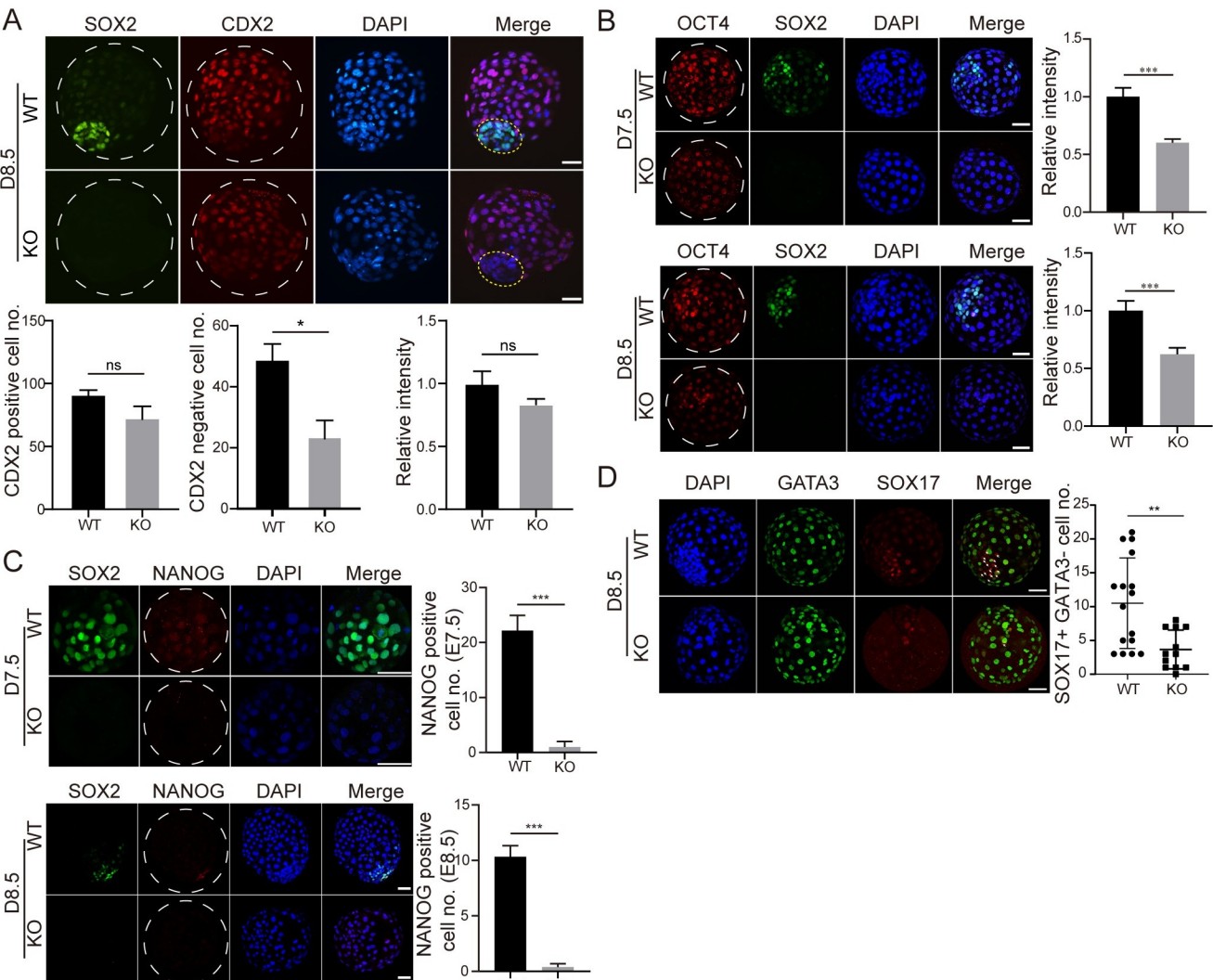

**Fig 5. SOX2 is indispensable for NANOG and OCT4 expression in the ICM of bovine blastocysts.** A. Immunostaining analysis of CDX2, a marker of trophectoderm (TE), in WT and *SOX2* KO blastocysts at D8.5. Bottom: Total cell counting analysis of TE cells (CDX2$^+$) in WT and *SOX2* KO blastocysts (Three replicates of 6–8 blastocysts per group were analyzed). B. Immunostaining analysis of OCT4, a pluripotency marker, in WT and *SOX2* KO blastocysts at D7.5 and D8.5 (Two independent replicates of 12–17 embryos per group were analyzed at D7.5 and D8.5, respectively). C. Immunostaining analysis of NANOG, an epiblast marker, in WT and *SOX2* KO blastocysts at D7.5 and D8.5 (Two independent replicates of 13–14 embryos per group were analyzed at D7.5 and D8.5, respectively). D: Immunostaining analysis of GATA3 (a marker for trophectoderm) and SOX17 (a marker for primitive endoderm) in WT and *SOX2* KO blastocysts (Two independent replicates of 6–9 embryos per group were analyzed at D8.5, respectively). Asterisks refer to significant differences (*:P < 0.05; **:P <0.05; ***: P<0.001). Scale bar = 50 μm.

## Discussion

How the earliest cell fate decisions are made is a fundamental question due to their importance for the establishment of pregnancy and fetal development in mammals. Recent studies suggest species-specific regulations of these events. Here, we present the proof-of-evidence of base-editor-mediated gene knockouts with a robust efficiency in bovine early embryos. Using this platform, we find that SOX2 regulates OCT4 and NANOG expression and the disappearance of SOX2 in the TE of bovine blastocysts is dependent on CDX2. These results are different from those of mouse studies, highlighting a species-specific role and regulation of SOX2 in mammals.

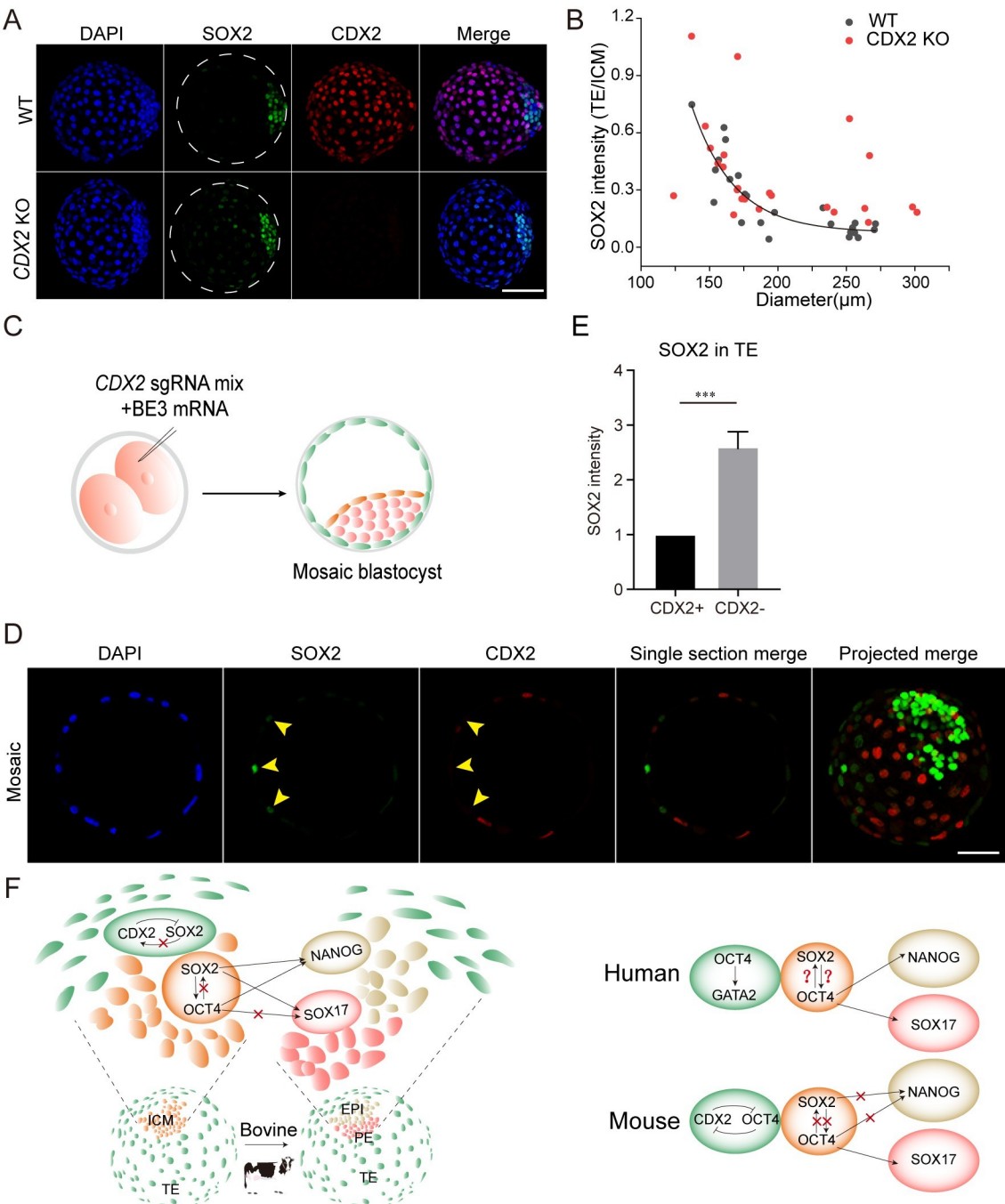

**Fig 6. CDX2 is required for the restricted expression of SOX2 in the ICM of bovine late blastocysts.** A. Immunostaining analysis of SOX2 in trophectoderm (TE) between WT and *CDX2* KO blastocysts. Scale bar = 50 μm. Seven replicates of 4–8 blastocysts per group. B: The correlation between SOX2 intensity (TE/ICM) and diameter in WT and *CDX2* KO blastocysts. C. Experimental scheme to produce CDX2 mosaic bovine blastocysts. D. Immunostaining analysis of SOX2 levels in CDX2⁻ TE cells relative to CDX2⁺ TE cells. Green: SOX2; Red: CDX2; Scale bar = 50 μm. E. Statistical analysis of SOX2 intensity in CDX2⁻ TE cells relative to CDX2⁺ TE cells. n = 12. Asterisks refer to significant differences (*:P < 0.05; **:P <0.05; ***: P<0.001). F. Summary and working model of functional relationship between SOX2 and other core lineage-specific genes in human, mouse and bovine blastocysts.

A longstanding barrier for studying gene functions in large animals is the lack of genetic tools to disrupt a gene of interest. The advent of CRISPR-Cas9 technology represents a powerful approach to achieve genome editing. However, recent studies indicate the use of Cas9 in human early embryos results in unintentional deletions of large fragments, raising concerns in addressing gene functional studies and clinical use [25,26]. We thus decided to use base editing system in the present study. Cytosine base editors are particularly useful for disrupting genes by introducing a premature stop codon into a gene of interest without creating double-strand breaks or indels. Here, our studies report that BE3 and ABE7.10 facilitated gene editing with an efficiency above 79% in bovine embryos. Importantly, we found no obvious off-target editing at potential sites, indicating the specific effects we documented in the present study. To maximize the editing efficiency, we microinjected 2 or 3 sgRNAs together and found the target gene can be deleted completely in all blastomeres in around 80% embryos with only less than 10% embryos exhibiting mosaicism. We believe that this approach is a powerful tool to dissect gene functions and produce genome-edited cattle.

A series of transcription factors participate in the lineage development, including CDX2, SOX2, OCT4, and NANOG. These factors are originally identified as lineage-specific in mice and are also present in other mammals. Nonetheless, whether their functions are conserved across species remains poorly determined, especially in large animals. OCT4 expression lasts a long time in the TE of bovine blastocysts, in contrast to the expression pattern observed in mice [27,28]. OCT4 is restricted into the ICM later than SOX2 in bovine embryos, suggesting SOX2 is required first for the establishment of pluripotency. Consistently, *OCT4* KO does not affect the expression of SOX2 (Fig 1I and 1J), however, *SOX2* KO leads to reduced OCT4 expression in bovine blastocysts. Remarkably, we observed even no NANOG expression in both D7.5 and D8.5 *SOX2* KO blastocysts. A recent study demonstrates that OCT4 is required for NANOG expression in bovine blastocysts [6], suggesting that SOX2 may regulate NANOG indirectly through OCT4. However, we speculate that SOX2 also directly regulates the expression of NANOG because OCT4 is not completely lost when SOX2 is deleted (Fig 6G).

RNA-seq results reveal a large-scale disruption of the transcriptome upon SOX2 deletion in D7.5 blastocysts with 2074 genes affected. In comparison, a previous report has shown that only 472 genes are dysregulated in bovine D7 *OCT4*-null blastocysts [6], indicating the molecular consequence of *SOX2* KO is more severe than OCT4.

Mutual feedback between lineage-specific genes in mammalian embryos has been reported previously [1]. OCT4 and CDX2 are mutually regulated by each other in the ICM and TE in mouse blastocysts [1]. However, Sox2 is restricted to inside cells by a Cdx2-independent mechanism in mice [3]. Results herein show CDX2 is involved in suppressing SOX2 in the TE along with blastocyst expansion in cattle. This suppression is developmental context-dependent as it takes place when SOX2 becomes resctricted into the ICM but not earlier.

In conclusion, we demonstrate that the base editing system could be applied to bovine embryos. With this powerful tool, KO of three critical lineage-specific genes is successfully achieved in bovine embryos. Functional experiments prove that *SOX2* KO significantly disrupts OCT4 and NANOG expression. Meanwhile, the disappearance of SOX2 in the TE is dependent on CDX2. Altogether, our study reveals a species-specific role of SOX2 in the regulation of pluripotency and unique regulation of SOX2's restricted expression in bovine blastocysts.

## Materials and methods

### Materials

All chemicals and reagents were commercially obtained from Sigma (St. Louis, MO, USA) unless stated elsewhere.

## In vitro production of bovine embryos

Bovine embryos in vitro production, including in vitro maturation (IVM), in vitro fertilization (IVF) and in vitro culture (IVC) was performed as procedures published previously with slight modifications [29–31]. Briefly, cumulus-oocyte complexes (COCs) containing intact cumulus cells were collected from bovine ovaries obtained from a local abattoir. COCs were matured in Medium-199 (M4530) supplemented with 10% FBS (Gibco-BRL, Grand Island, NY), 1 IU/ml FSH (Sansheng Biological Technology, Ningbo, China), 0.1 IU/ml LH (Solarbio, Beijing, China), 1 mM sodium pyruvate (Thermo Fisher Scientific, Waltham, MA, USA), 2.5 mM GlutaMAX (Thermo Fisher Scientific, Waltham, MA, USA), and 10 μg/mL gentamicin at 38.5˚C under 5% $CO_2$ in humidified air for 22–24 hrs. COCs (60–100 per well in 4-well plates) were then incubated with spermatozoa ($1–5×10^6$) purified from frozen-thawed semen by using a percoll gradient in BO-IVF medium (IVF bioscience, Falmouth, Cornwall, UK). IVF condition was 38.5˚C under 5% $CO_2$ for 9–12 hrs. Putative zygotes were then removed of cumulus cells by pipetting up and down using Medium-199 (M7528) supplemented with 2% FBS (Gibco-BRL, Grand Island, NY, USA). Embryos were incubated in BO-IVC medium (IVF bioscience, Falmouth, Cornwall, UK) at 38.5˚C under 5% $CO_2$ in humidified air until use.

## sgRNA design, synthesis, and plasmid construction

BE-Designer online software (http://www.rgenome.net) was used to design sgRNAs. SgRNA sequences with appropriate GC content and low probability for off-target were selected to target the coding region of the gene of interest. The sticky end of BpiI: 5'-3' CACC and 5'-3' AAAC were added to the 5' ends of the sense and antisense strand, respectively (S1 Table). The DNA sequences were synthesized by Sangon Co., LTD (Shanghai, China). Then, sgRNA DNA oligos were annealed and cloned into a PX458 vector containing BpiI restriction sites with a T7 promoter.

## In vitro transcription

BE3 and ABE7.10 plasmids were purchased from Addgene (#73021 and #102919). After linearization with NotI, the plasmid underwent in vitro transcription using mMESSEAGE mMACHINE T7 kit (Invitrogen, Thermo Fisher Scientific, Waltham, MA, USA) and was purified by LiCl precipitation. sgRNAs were amplified and transcribed in vitro using MEGAshortscript T7 High Yield Transcription Kit (Invitrogen, Thermo Fisher Scientific, Waltham, MA, USA) according to manufacturer's instructions. Primers are listed in S2 Table. After transcription, sgRNAs were purified by ethanol precipitation.

## Microinjection of base editor mRNA

10–20 pL mixture of 100 ng/μL sgRNA and 200 ng/μL ABE7.10 or BE3 mRNA were microinjected into bovine zygotes at 12 h post insemination (hpi) by using a micromanipulator (TransferMan, Eppendorf, Germany). Control embryos were injected with the same amount of mRNA without sgRNA. To maximize the editing efficiency of the gene of interest, a cocktail of two or three sgRNAs was microinjected together with ABE7.10 or BE3 mRNA. Each sgRNA was kept at the same concentration (100 ng/μL). For constructing CDX2 mosaic embryos, we microinjected base editing components into one blastomere at 2-cell stage.

## Single bovine embryo PCR and genotyping

Injected embryos were collected at morula or blastocyst stage. Genomic DNA was isolated using an embryo lysis buffer (40 nM Tris-HCl, 1% Triton X-100, 1% NP-40 and 0.4 ng/mL

Proteinase K) at 55˚C for 1 h and 95˚C for 10 min. Nested PCR was performed and then amplicons were subject to Sanger sequencing. As for nested PCR, two rounds of PCR were performed by using primeSTAR HS DNA Polymerase (Takara, Cat. #R040A). PCR condition was: 98˚C for 2 min followed by 35 cycles of 98˚C for 10 s, 60˚C for 5 s, 72˚C for 1.5 min, and a final 5-min step at 72˚C. All primers used are listed in S3 Table.

### Targeted deep sequencing

Single embryo was subject to whole-genome amplification by using REPLI-g Mini Kits (QIA-GEN, Cat. No. 150023). The target sites and 6 potential off-target sites (S4 Table) that were predicted by an online software were amplified using PCR primers with barcode sequences (S5 Table). All amplicons were purified and subject to targeted deep sequencing.

### Immunofluorescence (IF)

Early embryos were rinsed three times with 0.1% PBS/PVP (polyvinylpyrrolidone), and fixed with 4% paraformaldehyde in PBS for 30 min, permeabilized with 0.5% Triton X-100/PBS for 30 min. Fixed samples were then blocked for 1–2 hrs with the buffer containing 10% FBS and 0.1% Triton X-100/PBS. Samples were incubated with primary antibodies for 2 hrs at room temperature or overnight at 4˚C. Then, embryos were treated with secondary antibodies for 2 hrs. Nuclear DNA was counterstained by DAPI for 15 min. Samples were mounted and observed with either an inverted epifluorescence microscope (Nikon, Chiyoda, Japan) or a Zeiss LSM880 confocal microscope system (Zeiss, Oberkochen, Germany). For confocal microscopy, Z-stacks were imaged with 5 μm intervals between optical sections. Stacks were projected by maximum intensity to display signals of all blastomeres in one image. All antibody information was shown in S6 Table.

### Single blastocyst RNA-seq and data analysis

Single early blastocysts from WT and KO group were collected on D7.5. The zona pellucida of blastocysts was discarded with 0.5% pronase E. The RNA-seq libraries were constructed according to Smart-seq2 procedures as previously described [32]. In brief, polyadenylated RNAs were captured and reverse transcribed with Oligo(dT) primer, then the cDNA was pre-amplified using KAPA HiFi HotStart ReadyMix (kk2601). Pre-amplified cDNA was purified with Ampure XP beads (1,1 ratio) and fragmented by Tn5 enzyme (Vazyme, TD502). PCR amplification for 15–18 cycles was performed to prepare sequencing libraries, which were subject to paired-end 150 bp sequencing on a NovaSeq (Illumina) platform by Novogene. The raw sequencing reads were trimmed with Trimmomatic (version 0.39) [33] to generate clean data and mapped to ARS-UCD1.2 with Hisat2 (version 2.1.0) [34]. The raw counts were calculated with featureCounts (version 1.6.3) [35] and underwent differential expression analysis using DESeq2 [36]. The differentially-expressed genes between WT and KO groups were identified using an FDR-adjusted P-value (Padj) $< = 0.05$. and Foldchange $> = 2$ or $< = 0.5$. FPKM for each sample was calculated with Cufflinks [37] for heatmap visualization, and heatmaps generated using pheatmap package in R. Gene ontology analysis was performed with the Database for Annotation, Visualization and Integrated Discovery (DAVID) [38,39]. All RNA-seq files are available from the Gene Expression Omnibus database (accession number GSE189318).

### Statistical analysis

All experiments were replicated at least three times unless stated. Two-tailed unpaired student t-tests were used to compare differences between two groups. One-Way ANOVA was used to

analyze significant differences among three groups. The fluorescent intensity was analyzed using Image J as described previously [29]. Briefly, the nuclear region was encircled based on the DAPI signal and the intensity measured. The same region was moved to the cytoplasm area and background intensity obtained. The specific signal was calculated by subtracting the cytoplasmic intensity from the nuclear intensity. Finally, the data were normalized to the relative channels in control groups. The graphs were constructed by GraphPad Prism 8.0 (GraphPad Software, USA). P<0.05 refers to statistical significance.

## Supporting information

**S1 Fig. Base editors enable efficient genome editing in bovine embryos.** A. Target sites of sgRNA designed for *SMAD4*, *TEAD4*, *CDX2*. Red letters represent the target sites of BE3 or ABE7.10. Blue letters represent protospacer adjacent motif (PAM) sequences. B. Experimental scheme for base editing in bovine early embryos. C and D: Results of targeted deep sequencing for base editing of *SMAD4* by BE3 in bovine embryos. 17 embryos were analyzed. E and F: Results of targeted deep sequencing for base editing of *SMAD4* by ABE7.10 in bovine embryos. 25 embryos were analyzed.
(TIF)

**S2 Fig. Analysis of off-target effects of base editor ABE7.10 and BE3.** A and B: Targeted deep sequencing analysis of 6 potential off-target sites for ABE7.10 (A) and BE3 (B).
(TIF)

**S3 Fig. Application of multi-gene base editing using ABE7.10 and BE3.** A and B: Embryonic developmental rate to reach morula stage in the WT and ABE7.10 (A) or BE3 (B) group (Two replicates of 13–18 embryos per group). C. Representative Sanger sequencing results of ABE7.10-meidated base editing. D. Representative Sanger sequencing results of BE3-mediated base editing. The red letters and frames represent the editing sites. The green letters represent PAM sequence. S-gRNA: *SMAD4* sgRNA; T-gRNA: *TEAD4* sgRNA; C-gRNA: *CDX2* sgRNA.
(TIF)

**S4 Fig. Single blastocyst RNA sequencing.** A. Single D7.5 early blastocysts were collected (n = 5 per group) to perform RNA sequencing. B. Early blastocysts collected on D7.5 and used for RNA-seq have similar total cell number. C. Validation of the genotypes of embryos used for RNA-sequencing in A. D. Principal component analysis (PCA) shows high correlation among samples in the same group. E and F. Principal component analysis (E) and hierarchical clustering analysis (F) by comparing the transcriptomes of wildtype (WT BL) and SOX2 KO (KO BL) D7.5 blastocysts with the one of wildtype D6 morula (WT MO, Datasets No.: GSE158679).
(TIF)

**S5 Fig. SOX2 knockout causes a reduction in NANOG and OCT4 level in D6.5 bovine morula.** A. Quantification analysis of the changes of OCT4 levels in TE and ICM as blastocyst expansion. TE: SOX2- cells; ICM: SOX2+ cells. B. The dynamics of NANOG and OCT4 expression accompanied by the blastocyst expansion. Green: NANOG; Red: OCT4 (Two replicates of 3–5 blastocysts at different stages). Scale bar = 50 μm C. Immunostaining pictures of NANOG and OCT4 in WT and SOX2 KO (KO) embryos at E6.5. D and E. Relative intensity of NANOG (D) and OCT4 (E) in WT and KO groups. Scale bar = 50 μm.
(TIF)

**S6 Fig. CDX2 KO did not affect bovine early embryonic development.** A. sgRNAs designed to target *CDX2*. B. Representative Sanger sequencing results of *CDX2* editing. The red letters

represent editing sites. C. Immunostaining detection of CDX2 (Three replicates of 5–8 blastocysts per group). Red: CDX2. Scale bar = 50 μm. D. Editing types analysis using *CDX2* sgRNA1, sgRNA2 and sgRNA3 (61 embryos were detected). E and F. *CDX2* KO has no effect on the rate of blastocyst formation (Ten replicates of 20–25 embryos per group). Scale bar = 100 μm.

(TIF)

**S1 Table. The synthesis of sgRNAs sequence.**
(PDF)

**S2 Table. The primers information of sgRNA template for in vitro transcription.**
(PDF)

**S3 Table. Nested PCR primer sequences for preparing Sanger sequencing samples.**
(PDF)

**S4 Table. The predicted potential 6 off-target sites for *SMAD4*.**
(PDF)

**S5 Table. PCR primer sequences for preparing targeted next generation sequencing samples.**
(PDF)

**S6 Table. Antibody information.**
(PDF)

## Acknowledgments

We thank all members of the K. Zhang laboratory for their helpful discussions.

## Author Contributions

**Conceptualization:** Kun Zhang.

**Formal analysis:** Yanna Dang.

**Investigation:** Lei Luo, Yan Shi, Yanna Dang, Shuang Li.

**Methodology:** Huanan Wang, Zizengchen Wang, Shaohua Wang.

**Resources:** Huanan Wang.

**Supervision:** Shaohua Wang, Kun Zhang.

**Writing – original draft:** Lei Luo.

**Writing – review & editing:** Yan Shi, Kun Zhang.

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
