## [Decision Letter · Decision Letter 0]

10 Feb 2022

Dear Dr Zhang,

Thank you very much for submitting your Research Article entitled 'Base editing in bovine embryos reveals a species-specific role of SOX2 in regulation of pluripotency' to PLOS Genetics.

The manuscript was fully evaluated at the editorial level and by independent peer reviewers. The reviewers are quite enthusiastic about this important problem, but two of the reviewers raised some substantial concerns about the current manuscript. Importantly, they would like to see the RNA-seq analysis improved as well as significantly improved immunofluorescence to provide more convincing data on the downregulation of Oct4. Based on the reviews, we will not be able to accept this version of the manuscript, but we would be willing to review a much-revised version. We cannot, of course, promise publication at that time.

If you decide to revise the manuscript for further consideration at PLOS Genetics, please aim to resubmit within the next 60 days, unless it will take extra time to address the concerns of the reviewers, in which case we would appreciate an expected resubmission date by email to plosgenetics@plos.org.

[LINK]

We are sorry that we cannot be more positive about your manuscript at this stage. Please do not hesitate to contact us if you have any concerns or questions.

Yours sincerely,

Marisa S Bartolomei

Associate Editor

PLOS Genetics

Gregory Barsh

Editor-in-Chief

PLOS Genetics

Reviewer's Responses to Questions

**Comments to the Authors:**

Reviewer #1: This manuscript from K Zhang’s laboratory represents an excellent contribution to the field of early mammalian development and is highly commendable in not only rigorously adapting base editing to gene knock-outs in a difficult-to-study mammal, but also in using the methodology to knock-out three of the genes shown to be important in mice preimplantation development. The study highlights important differences in the drivers of preimplantation development between mice and cattle which makes this work very interesting and important.

There are though some minor but rather serious mistakes and a small part of the data is unconvincing.

In the abstract the authors say SOX2 is dispensable for OCT4 and NANOG but mean indispensable!!

In the section “Base editors enable…”, it would be worthwhile to also indicate the % editing for each of the three genes for both methods.

The section on OCT4 knock-out should have the supplementary figure moved into the main paper. Why were the shown FigS3I and J panels not discussed in the text? It appears that Sox2 protein is upregulated/maintained upon OCT4 knock-out. This is quite important data.

The section on Sox2 expression in bovine early embryos should probably include a reference to ref.6 as this is not all that novel.

The incorrect figure is quoted in line 155. Also, it was not shown in Fig 5A that the number of ICM cells was decreased! This data needs to be added.

The immunofluorescence of Figure 5 is in some instances unconvincing, even when viewed at magnification on a screen. In particular the highly significant down regulation of OCT4 protein at E7.5 is difficult to believe. Maybe the authors can linearly adjust the histograms on the confocal (equally for KO and Control) to allow better visualisation. Indeed the CDX2 intensity reduction in Fig 5A looks more real but is claimed to be non-significant. In Fig 5C NANOG staining is so weak even in controls as to be deemed background, making the reduction unconvincing.

The statement in lines 185-7 is too sweeping. These results do not substantiate that SOX2 is necessarily upstream of OCT4 and NANOG. If loss of SOX2 wipes out the ICM cells as claimed this will also remove NANOG and OCT4 expression in these cells, whether SOX2 protein controls their expression or not.

The CDX2 mosaic experiment is lovely, well done, and also well interpreted in line 254, that the suppression is developmental context-dependent.

In the methods section, please provide details of microinjector apparatus and procedure. When coinjection several sgRNA’s indicate whether each was kept at the same concentration. For genotyping, was this done after immunofluorescence and how were mosaic embryos identified (Fig 3E etc)? Indicate nested PCR conditions.

I would also put something in the abstract to indicate that you have knocked-out three genes with your methodology.

Minor

Line 46 “by the” instead of “of the”

Line 54, references missing

Line 126: greatly

Line 146: codon

Line 155: word order, use “decreased rather dramatically”

Reviewer #2: This manuscript by Luo and colleagues examines the regulation of gene expression in early bovine embryos. Generally speaking, the study of non-rodent mammalian development is significant because developmental mechanisms can differ between rodents and humans. By contrast, some developmental mechanisms may be more similar between cow and human during preimplantation, thus meriting the study of bovine preimplantation gene regulatory mechanisms.

The authors use CRISPR-mediated base-editing to investigate the roles and regulation of SOX2 in the cow embryo. Published studies have shown that SOX2 is an early pluripotency marker in mouse, but not in human or cow. Overall, the novelty of the study is not as high as desired because other groups have characterized the expression pattern of SOX2 in cow preimplantation embryos (Gerri et al., Nature 2020) and have evaluated the consequences of Sox2 knockout in this same context (Simmet et al., PNAS).

Some novelty is provided by 1) use of base editing (as opposed to reliance on CRISPR-induced indels) in cow embryos, 2) RNA-seq analysis of Sox2 knockout embryos, and 3) evaluation of CDX2 as an upstream regulator of the Sox2 expression pattern. However, not all of these directions are supported by appropriately rigorous study. As detailed below, the RNA-seq analysis pipeline is superficial and error-prone; the Cdx2 phenotype is weak (very modest affects on Sox2 expression).

Admittedly, additional novelty is provided by the observation that Sox17 expression is lowered in the absence of Sox2, indicative of either a) a novel cell-autonomous role for Sox2 in promoting expression of Sox17 or b) a conserved non cell-autonomous role for Sox2 in promoting expression of Sox17. However, these mechanistic possibilities are not explored. All in all, the manuscript falls short of making a big splash.

Detailed comments

Figure 1 could be moved to supplemental or cut entirely. This level of analysis is only really relevant to the study of the gene knockouts studied in this paper (Sox2 and Cdx2) – rather than Smad4, Tead4, and Cdx2).

Figure 2 is redundant with data already described by Gerri et al., Nature 2020

Figure 3 includes multiple pairwise comparisons (t-tests for 3 samples at a time) – leading one to question the reported significance.

Figure 4 also relies on t-tests for RNA-seq data, when false discovery rate is the standard. Additionally, the RNA-seq data do not take into account whether the Sox2 ko embryos could be developmentally delayed and whether this could explain the reported gene expression differences.

Figure 5 – why is the CDX2 level reduced in the SOX2 KO in panel A? Why isn’t OCT4 detectable in WT panel B or NANOG detectable in WT panel C? How was the relative intensity quantified?

Figure 6 – Panel A is attempting to show that CDX2 is responsible for repressing expression of SOX2 in the trophectoderm, but SOX2 is barely visible in a few CDX2 KO trophectoderm cells (likewise for panel E). In panels B, C, the SOX2 expression level in WT and CDX2 KO is quantified in terms of intensity – was this measured for all trophectoderm cells? A subset?

Reviewer #3: This is an important paper on comparative developmental biology of mammalian embryos. The results are important and supported by the experimental data. The paper rises to the level of importance expected for PLOS Genetics because of its importance to understanding how fundamental mechanisms of development have diverged between species.

All of my comments are very minor. The paper is well written but could use some editing for English. Moreover, the title and abstract does not do justice to the paper since little is said about the impact of the OCT4 knockout and the abstract does not make clear why CDX2 is believed to regulate SOX2 expression.

Here are the minor comments:

Line 40 (remarkably) spelling

Line 46 – reword this sentence. Perhaps “begins with the first sequential cell fate decisions occurring in close temporal relationship to each other”.

Line 80 – change to “highly-efficient”

Line 115 – change to “tested”

Line 133 – change to “be expressed”

Line 196 – change to “out of”

Methods – don’t capitalize names of chemicals (sodium pyruvate etc.)

L 310 – no space between degree symbol and number

Tables 1 and 2 – italicize gene symbols

**Have all data underlying the figures and results presented in the manuscript been provided?**

Reviewer #1: Yes

Reviewer #2: Yes

Reviewer #3: Yes

PLOS authors have the option to publish the peer review history of their article (what does this mean?). If published, this will include your full peer review and any attached files.

Reviewer #1: No

Reviewer #2: No

Reviewer #3: No

---

## [Decision Letter · Decision Letter 1]

19 Apr 2022

Dear Dr Zhang,

Thank you very much for submitting your Research Article entitled 'Base editing in bovine embryos reveals a species-specific role of SOX2 in regulation of pluripotency' to PLOS Genetics.

The manuscript was fully evaluated at the editorial level and by the same reviewers as the first round. While two of the reviewers were satisfied with the revision, one reviewer is still concerned with staging and the potential for the mutations to result in embryonic delay, which compromises the interpretation of the phenotype. Based on the reviews, we will not be able to accept this version of the manuscript, but we would be willing to review a revised paper if you can address all of the concerns of the reviewer. 

If you decide to revise the manuscript for further consideration at PLOS Genetics, please aim to resubmit within the next 60 days, unless it will take extra time to address the concerns of the reviewers, in which case we would appreciate an expected resubmission date by email to plosgenetics@plos.org.

[LINK]

We are sorry that we cannot be more positive about your manuscript at this stage. Please do not hesitate to contact us if you have any concerns or questions.

Yours sincerely,

Marisa S Bartolomei

Associate Editor

PLOS Genetics

Gregory Barsh

Editor-in-Chief

PLOS Genetics

Reviewer's Responses to Questions

**Comments to the Authors:**

Reviewer #1: I thank the authors for answering all my queries well and presenting additional evidence for the IF results.

Reviewer #2: I have reviewed the authors’ rebuttal, but I still have concerns about the revised manuscript, which I delineate below. I consider these to be major concerns, because they pertain to the paper’s main big points. Therefore, I am not confident in the quality of this study in its current form.

• As I mentioned in the first round of review, the SOX2 time course was already shown in Gerri et al., 2020, Extended Data Fig. 5C: https://www.ncbi.nlm.nih.gov/pmc/articles/PMC7116563/figure/F8/

The authors argue that they show more time points than Gerri et al., which is true. However, those additional time points encompass stages when SOX2 is not expressed (prior to 8-cell stage). Therefore, this aspect of the manuscript is a more incremental contribution to our understanding of bovine preimplantation.

• I previously raised concerns about embryo staging. Throughout the paper, the authors are inconsistent in their definitions of stage, sometimes reporting cell counts, but usually relying on surrogates for determining stage, such as time in culture (e.g., E7.5, E8.5), embryo size, or subjective terms (e.g., morula, early blastocyst, late blastocyst). This issue is not resolved, and complicates interpretation of phenotypes, detailed below.

• CDX2 KO phenotype: does CDX2 repress SOX2 in the TE? In the first round of review, I raised the concern that the CDX2 phenotype could be a result of developmental delay. Since the CDX2 phenotype is the perdurance of SOX2 in the TE slightly longer than what is observed normally. The authors show nicely in a wild type time course in Fig. 2C,D that SOX2 is normally gradually diminished in the TE. They should show their data in this way for the CDX2 KO embryos.

Therefore, I cannot tell if the CDX2 KO and WT embryos are stage-matched and whether the reported phenotype (SOX2 perdurance) is “real.”

• Same topic, different KO: the authors show in Fig. 3G,H that total cell numbers are reduced in SOX2 KO embryos (by half, which is a pretty dramatic difference). Is this a developmental delay or are cells dying? Do embryos have the same number of inside and outside cells? The authors look at the number of CDX2-positive and -negative cells (Fig. 5A), but this is gene expression, and not lineage allocation per se. This raises questions as to whether E7.5 WT and KO embryos are appropriate comparisons for the RNAseq or immunofluorescence experiments shown in Fig. 4-5 and conclusions drawn about regulation of OCT4 and NANOG expression.

Rather, the authors argue that the embryos are the same stage because they are the same size (Fig. S5C). But size is not a reliable indicator of developmental stage during preimplantation, when cleavage divisions and the dynamic and iterative process of blastocoel expansion complicate interpretation of embryo size. I appreciate Fig. R3, as an indicator that they can resolve transcriptional differences between morula and blastocyst, but I can’t tell if this is sufficiently granular to address my concerns. In addition, these data do not serve as a good control if they are not included in the revised manuscript.

Reviewer #3: This is a very good and important paper.

**Have all data underlying the figures and results presented in the manuscript been provided?**

Reviewer #1: Yes

Reviewer #2: **No: **Not clear where the RNA-seq data are posted

Reviewer #3: None

PLOS authors have the option to publish the peer review history of their article (what does this mean?). If published, this will include your full peer review and any attached files.

Reviewer #1: No

Reviewer #2: No

Reviewer #3: No

---

## [Editor Report · Decision Letter 2]

22 Jun 2022

Dear Dr Zhang,

We are pleased to inform you that your manuscript entitled "Base editing in bovine embryos reveals a species-specific role of SOX2 in regulation of pluripotency" has been editorially accepted for publication in PLOS Genetics. Congratulations!

Yours sincerely,

Marisa S Bartolomei

Associate Editor

PLOS Genetics

Gregory Barsh

Editor-in-Chief

PLOS Genetics

Comments from the reviewers (if applicable):

**Data Deposition**

http://datadryad.org/submit?journalID=pgenetics&manu=PGENETICS-D-22-00012R2

**Press Queries**

---

## [Editor Report · Acceptance letter]

29 Jun 2022

PGENETICS-D-22-00012R2 

Base editing in bovine embryos reveals a species-specific role of SOX2 in regulation of pluripotency 

Dear Dr Zhang, 

We are pleased to inform you that your manuscript entitled "Base editing in bovine embryos reveals a species-specific role of SOX2 in regulation of pluripotency" has been formally accepted for publication in PLOS Genetics! Your manuscript is now with our production department and you will be notified of the publication date in due course.

With kind regards,

Zsofia Freund

PLOS Genetics

On behalf of:
